# The Interactive Effects of Severe Vitamin D Deficiency and Iodine Nutrition Status on the Risk of Thyroid Disorder in Pregnant Women

**DOI:** 10.3390/nu14214484

**Published:** 2022-10-25

**Authors:** Wei Lu, Zhengyuan Wang, Zhuo Sun, Zehuan Shi, Qi Song, Xueying Cui, Liping Shen, Mengying Qu, Shupeng Mai, Jiajie Zang

**Affiliations:** Division of Health Risk Factors Monitoring and Control, Shanghai Municipal Center for Disease Control and Prevention, Shanghai 200336, China

**Keywords:** vitamin D deficiency, thyrotropin receptor antibodies, pregnancy, iodine

## Abstract

Thyroid dysfunction is associated with both vitamin D deficiency and iodine; however, it is unclear whether they interact. This study aimed to investigate whether and to what extent the interactions between vitamin D and iodine contribute to the risk of thyroid disorder. Participants (*n* = 4280) were chosen using multistage, stratified random sampling from Shanghai. Fasting blood was drawn for the 25(OH)D and thyroid parameter tests. Spot urine samples were gathered to test for urine iodine. To evaluate the interactive effects of vitamin D and iodine, crossover analysis was carried out. Pregnant women with a high urinary iodine concentration (UIC) and severe vitamin D deficiency had a significantly higher risk of thyrotropin receptor antibody (TrAb) positivity (odds ratio = 2.62, 95% confidence interval (CI): 1.32, 5.22) in the first trimester. Severe vitamin D deficiency and high UIC interacted positively for the risk of TrAb positivity (relative excess risk due to interaction = 1.910, 95%CI: 0.054, 3.766; attributable proportion = 0.700, 95%CI: 0.367, 1.03). Severe vitamin D deficiency combined with excess iodine could increase the risk of TrAb positivity in pregnant women in the first trimester.

## 1. Introduction

The thyroid status of pregnant women is subject to substantial pregnancy-related physiological changes. The steep increase in chorionic gonadotropin (hCG) levels during the first trimester may result in an increased production of thyroid hormones, making serum thyrotropin (TSH) levels decrease in the first trimester and increase in the second and third trimesters. Total thyroxine (TT4) and total triiodothyronine (TT3) levels increase by 50% during pregnancy owing to a 50% increase in thyroxine-binding globulin levels [1,2]. Thyroid-related pathophysiologic changes may be aggravated by pregnancy, and some obstetric conditions, such as gestational trophoblastic disease or hyperemesis gravidarum, may also affect thyroid gland function [3,4]. Recent research has revealed a connection between thyroid malfunction and vitamin D deficiency in pregnant women [5,6]. This finding is interesting, because both vitamin D deficiency and thyroid dysfunction are a matter of concern to women during pregnancy. Almost one in five pregnant women and one in three newborns suffer from vitamin D deficiency [7]; overt hypothyroidism occurs in 0.3–0.5% of pregnancies; subclinical hypothyroidism occurs in 2–3%; thyroid autoantibodies are found in 5–15% of women during pregnancy [8].

Autoimmune thyroid diseases (AITDs) are the most prevalent autoimmune diseases during pregnancy. Studies have reported lower serum vitamin D levels in patients with AITD compared with healthy controls [9,10]. However, other studies have reported no significant association between serum vitamin D levels and thyroid autoimmunity [11,12]. Although the results were adjusted for possible confounding factors, such as age, body mass index (BMI), and current smoking status in those studies, the nutritional status of iodine was not included in most of them. A study in Korea showed that the association between thyroid dysfunction and vitamin D was affected by different iodine status, suggesting a potential interaction effect of vitamin D status and iodine nutrition status on thyroid function [13].

Hence, the main purpose of this study was to evaluate the interactive effects of vitamin D status and iodine nutrition status on the risk of thyroid disorder in pregnant women.

## 2. Materials and Methods

### 2.1. Study Population

This was a prospective population-based birth cohort study, which was part of the Iodine Status in Pregnancy and Offspring Health Cohort (ISPOHC) study. We followed pregnant women from pregnancy to fetal delivery from 2017 to 2018. The inclusion criteria comprised: resident pregnant women who were registered as part of the population or who had lived in Shanghai for more than 6 months and were registered in health service centers of Shanghai. The exclusion criteria comprised: women suffering from serious mental disorders, hepatitis (infectious stage), active tuberculosis, acquired immunodeficiency syndrome (AIDS), and those with a history of thyroidectomy or other endocrine diseases. The study was approved by the Ethical Committee of the Shanghai Municipal Center for Disease Control and Prevention (EC No. 2017/13, approval date: 25 April 2017). All surveys were conducted after obtaining written consent from the respondents.

### 2.2. Basic Characteristics, Food Consumption, and the Health Status of Pregnant Women

All the basic information was recorded using a questionnaire, including sex, age, drinking and smoking status, education level, and annual family income. The consumption frequency and the consumption of various kinds of food by pregnant women were collected using a well-designed food frequency questionnaire (FFQ). Professionally trained investigators helped the pregnant women complete the questionnaires through face-to-face interviews. The height and weight of the subjects were measured with tools of uniform brands and models to determine their BMI.

### 2.3. Thyroid Gland Function Assessment

Venous blood (5 mL) obtained from the women was centrifuged at 1800× *g* for 10 min. After that, serum was separated and stored at −80 °C. Antibody levels against serum thyrotropin (TSH), thyroid peroxidase, thyroglobulin, and thyrotrophic receptor were measured using a Cobas Elecsys 602 (Roche Diagnostics, Basel, Switzerland) system. We detected serum levels of TSH, free triiodothyronine (FT3), free thyroxine (FT4), total triiodothyronine (TT3), and total thyroxine (TT4) using an automatic luminescent immune analyzer (Cobas e 602; Roche). Hypothyroidism was defined as serum TSH concentration during pregnancy >2.5 mU/L in the first trimester and >3.0 mU/L in the second and third trimesters [14]. Thyroid peroxidase antibody (TPOAb) positivity was defined as TPOAb ≥ 34 IU/mL, and thyroglobulin antibody (TgAb) positivity was defined as TgAb ≥ 115 IU/mL [15]. Thyrotrophic receptor antibody (TrAb) positivity was defined as TrAb ≥ 1.75 IU/L [16].

### 2.4. Classification of Vitamin D Status

The levels of 25-hydroxy vitamin D(25(OH)D) were evaluated using the electrochemiluminescence (ECL) technique. A vitamin D level ≥ 30 ng/mL was defined as adequate; a level of 21–29 ng/mL was defined as insufficiency; a level of 12–20 ng/mL was defined as deficiency; a level < 12 ng/mL was defined as severe deficiency, according to the International Association of Endocrinology [17].

### 2.5. Assessment of Iodine Nutrition Status

Urine collection and the iodine urinary test method were carried out according to the methods detailed in our published articles [18]. Iodine status was categorized as follows: insufficiency (urinary iodine concentration (UIC) less than 150 μg/L) (low UIC); adequate (UIC more than 150 μg/L and less than 249 μg/L); above requirements (UIC more than 250 μg/L) (high UIC) [19].

### 2.6. Statistical Analyses

The modules of statsmodels and scipy.stats in python 3.6 were used for all data cleaning and analysis. Continuous variables are expressed as the means ± the standard errors. Independent group differences were examined using the Mann-Whitney U test for comparison of two groups, and the Kruskal-Wallis test was used for the comparison of categories with more than two groups. The interactions between severe vitamin D deficiency and iodine status on the risk of thyroid disorder were evaluated by crossover analysis [20,21]. If there is no biological interaction, relative excess risk due to interaction (RERI) and attributable proportion due to interaction (AP) are equal to 0. A *p*-value < 0.05 was considered statistically significant in all tests, unless otherwise noted.

## 3. Results

### 3.1. Vitamin D Status of Pregnant Women in Shanghai

A total of 4280 pregnant women were included in the study. Among them, 12.88% were suffering from severe vitamin D deficiency; only 10.63% had adequate vitamin D. The average serum 25(OH)D concentrations in the first trimester, second trimester, and third trimester were 19.57 ± 7.05, 20.06 ± 7.36, and 19.75 ± 7.16 ng/mL, respectively. Correspondingly, 65.32%, 56.70%, and 50.31% of the pregnant women suffered from vitamin D deficiency in the first trimester, second trimester, and third trimesters (Table 1). During pregnancy, the 25(OH)D concentration increased with gestational age, and the vitamin D level of most pregnant women was less than 20 ng/mL in this study (Figure 1A). No significant changes in the vitamin D level were observed among different UIC groups (Figure 1B).

### 3.2. Thyroid Function of Pregnant Women under Different Vitamin D Statuses

As shown in Table 2, women with severe vitamin D deficiency had lower FT4 levels compared with women with better vitamin status in the first trimester; women with severe vitamin D deficiency had higher TPOAb levels compared with women with better vitamin status in the third trimester. No significant differences in the levels of other thyroid hormones and antibodies were observed under different vitamin D statuses.

### 3.3. Interaction between Low UIC/High UIC and Severe Vitamin D Deficiency

We analyzed the interactions among low UIC/high UIC and vitamin D deficiency on the risk of hypothyroidism and TgAb, TPOAb, and TrAb positivity. All analyses were compared with adjustment for age, pre-pregnancy BMI, annual household income, educational level, gestation week, season, drinking status, smoking status, and vitamin D intake. A negative interaction was found between severe vitamin D deficiency and low UIC on the risk of hypothyroidism/TgAb/TPOAb/TrAb positivity (Table 3). Pregnant women with high UIC and severe vitamin D deficiency had a significantly higher risk for TrAb positivity (odds ratio (OR) = 1.86, 95%CI: 1.12, 2.91, *p* = 0.0069). Within the category of high UIC, women with severe vitamin D deficiency had a significantly higher risk of TrAb positivity (OR = 2.06, 95%CI: 1.31, 3.26, *p* = 0.0019). Within the category of low vitamin level, women with high UIC had significantly higher risk of TrAb positivity (OR = 2.10, 95%CI: 1.15, 3.83, *p* = 0.0157). A positive interaction was found between severe vitamin D deficiency and high UIC on the risk of TrAb positivity (RERI = 1.027, 95%CI: 0.162, 1.891; attributable proportion AP = 0.559, 95%CI: 0.258, 0.861) (Table 4).

We also analyzed the interactions according to different trimesters of pregnancy. The results are shown in the Appendix A. In the first trimester, pregnant women with high UIC and severe vitamin D deficiency had a significantly higher risk for TrAb positivity (OR = 2.62, 95%CI: 1.32, 5.22, *p* = 0.0060). Within the category of high UIC, women with severe vitamin D deficiency had a significantly higher risk of TrAb positivity (OR = 2.66, 95%CI: 1.33, 5.32, *p* = 0.0055). Within the category of low vitamin level, women with high UIC had a significantly higher risk of TrAb positivity (OR = 3.55, 95%CI: 1.32, 9.54, *p* = 0.0121). A positive interaction was found between severe vitamin D deficiency and high UIC on the risk of TrAb positivity (RERI = 1.910, 95%CI = 0.054, 3.766; AP = 0.700, 95%CI = 0.367, 1.032) (Appendix A). During other trimesters, a negative interaction was found between severe vitamin D deficiency and low UIC/high UIC on the risk of hypothyroidism/TgAb/TPOAb/TrAb positivity (Appendix A).

### 3.4. Thyroid Hormone Levels of Pregnant Women under Different TrAb Statuses

As shown in Table 5, serum TSH levels were decreased in women who were TrAb positive compared with those in women who were TrAb negative in the third trimester. Total triiodothyronine was decreased in women who were TrAb positive compared with that in women who were TrAb negative in the first trimester.

Free triiodothyronine was decreased in women who were TrAb positive compared with that in women who were TrAb negative in the second and third trimesters. Total thyroxine was increased in women who were TrAb positive compared with that in women who were TrAb negative in all trimesters, and free thyroxine was increased in the second and third trimesters.

## 4. Discussion

The mean 25(OH)D concentration was 19.80 ± 7.19 ng/mL, and 55.53% of pregnant women had vitamin D deficiency in Shanghai. Severe vitamin D deficiency combined with excess iodine could increase the risk of TrAb positivity in pregnant women in the first trimester.

Vitamin D deficiency/insufficiency during pregnancy can cause a series of adverse outcomes for pregnant women and their offspring. The global prevalence of vitamin D deficiency is 54% among pregnant women and 75% among newborns. The prevalence of vitamin D deficiency and severe deficiency in pregnant women is: Americas (64%, 9%), European (57%, 23%), Eastern Mediterranean (46%, 79%), Southeast Asian (87%, not available), and Western Pacific (83%, 13%) [22]. The deficiency rate in our study was lower than the rate reported ten years ago in Shanghai [23]. During pregnancy, the 25(OH)D concentration increased with gestational age, and women in the first trimester had the highest prevalence of vitamin D deficiency and severe deficiency in the pregnant period. Early pregnancy is crucial for the development of embryonic organs. Unfavorable newborn outcomes were independently attributable to low vitamin D status in early pregnancy [24]. Fetal growth, low exposure to sunlight, and low dietary intake are the cause for vitamin D deficiency [25]. Supplementation of vitamin D was associated with increased circulating 25(OH)D levels [26] and decreased the risk of endocrine diseases, such as thyroid dysfunction [27] and gestational diabetes mellitus [28]. Therefore, monitoring of 25(OH)D levels should be used to detect vitamin D deficiency in early pregnancy, and supplements should be given as needed.

Recently, a compelling body of experimental data suggested that vitamin D regulates the innate and adaptive immune systems on a fundamental level [29]. Vitamin D is regarded as one of the natural immune modulators and a regulator of several immune-mediated processes, in addition to its roles in calcium/phosphate homeostasis [30]. Some studies have found that vitamin D deficiency is related to thyroid disorder in pregnant women [5,31]. Nevertheless, there have also been some reports contradicting such a relationship [12,32], making it challenging to come to a consensus regarding the relationship between pregnant women’s thyroid function and vitamin D status. Possible confounding factors, such as age, BMI, and current smoking status, were adjusted in those studies when studying the association between vitamin D deficiency and thyroid disorder. However, the nutritional status of iodine was not included in most studies. Both iodine deficiency and excess may increase the risk of developing a thyroid disorder [4]. In our study, women with severe vitamin D deficiency had significantly higher risk of TrAb positivity within the category of high UIC. However, an association was not found within the category of low UIC, suggesting that the relationship between vitamin D deficiency and thyroid disorder may be influenced by different iodine nutritional statuses.

To our knowledge, few studies have focused on the effect of the interaction between vitamin D and iodine status on thyroid disorder. There was a significant interaction between high UIC and severe vitamin D deficiency on TrAb positivity in the first trimester (RERI = 1.910, 95%CI: 0.054, 3.766) in our study, suggesting that there would be a 1.910 relative excess risk caused by the additive interaction. The AP was 0.700, indicating that, when exposed to both risk factors, 70.00% of the TrAb positivity was attributable to the additive interaction. In addition to its role in calcium homeostasis and bone formation, a modulatory role of the active form of vitamin D on cells of the immune system has been described [33]. The receptors of vitamin D are the same as thyroid hormones, and any modification in an individual’s constructor genes is prone to thyroid autoimmune diseases [34]. Iodine excess may also trigger the development of autoimmune thyroid disease such as lymphocytic Hashimoto’s thyroiditis [35]. However, it is still unclear how severe vitamin D deficiency combined with excess iodine influences thyroid function. The mechanism may refer to the TSH receptor-adenylate cyclase system. Adenosine 3′5′-cyclic monophosphate (cAMP) is involved in a negative control of tissue growth [36] and deiodinase function. Deiodinase II expression in the human thyroid gland is regulated at the transcriptional level through the TSH receptor-G-cAMP regulatory cascade, which may be related to the increase in the circulating T3 level in patients with Graves’ disease and hyperfunctioning thyroid adenoma [37]. It is reported that 1,25-(OH)_2_D_3_ is specifically bound to the vitamin D receptor [38] and was shown to inhibit the TSH/cAMP stimulatory pathway at several steps in a rat thyroid cell line [39]. High concentrations of iodide also inhibited cyclic AMP accumulation by a process that was prevented by inhibitors of iodide trapping (perchlorate) and of iodide oxidation (methimazole) [40]. Further studies are required to determine the effects of vitamin D deficiency and excess iodine intake co-exposure on the TSH receptor-adenylate cyclase system.

Grave’ disease (GD) is the most frequent cause of hyperthyroidism during pregnancy. The prevalence rate of overt hyperthyroidism is approximately 0.1–0.4%, and GD accounts for 85–90% of all cases among pregnant women [41]. In our study, total thyroxine was increased in women who were TrAb positive compared with its level in women who were TrAb negative in all trimesters. Moreover, free thyroxine was increased in the second, as well as the third trimester. The thyrotropin receptor is a major target for autoantibodies in GD. Stimulating TrAb activates the thyrotropin receptor, resulting in overproduction of thyroid hormone [42].

## 5. Conclusions

In conclusion, vitamin D deficiency was prevalent in pregnant women in Shanghai, and there was an additive interaction of excess iodine and vitamin D deficiency on TrAb positivity in pregnant women in the first trimester.

## Figures and Tables

**Figure 1 nutrients-14-04484-f001:**
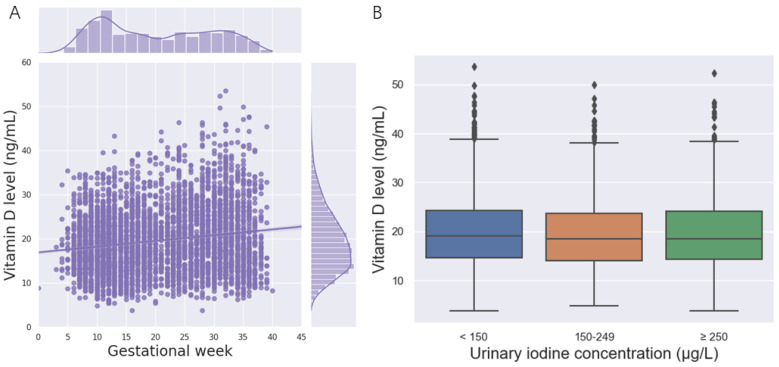
(25(OH)D) concentration of the participants. (**A**)The relationship between the 25(OH)D concentration and gestational week. (**B**) The relationship between the 25(OH)D concentration and the urinary iodine concentration.

**Table 1 nutrients-14-04484-t001:** General characteristics and vitamin D status of the participants.

	The First Trimester	The Second Trimester	The Third Trimester	All Trimesters
Vitamin D level (ng/mL)	19.57 ± 7.05	20.06 ± 7.36	19.75 ± 7.16	19.80 ± 7.19
Vitamin status				
Severe deficiency	196 (13.68)	200 (13.14)	155 (11.71)	551 (12.88)
Deficiency	740 (51.64)	663 (43.56)	511 (38.60)	1914 (44.73)
sufficiency	430 (30.01)	504 (33.11)	425 (32.10)	1359 (31.76)
Adequate	67 (4.68)	155 (10.18)	233 (17.60)	455 (10.63)
Age (years)				
<30	756 (52.76)	905 (59.46)	772 (58.30)	2433 (56.86)
30–35	467 (32.59)	432 (28.38)	380 (28.70)	1279 (29.89)
≥35	210 (14.66)	185 (12.16)	172 (12.99)	567 (13.25)
Annual household income (10,000 RMB)				
<10	198 (13.82)	289 (18.99)	283 (21.38)	770 (18.00)
10–20	598 (41.73)	654 (42.97)	531 (40.11)	1783 (41.67)
20–35	452 (31.54)	405 (26.61)	384 (29.00)	1241 (29.00)
≥35	185 (12.91)	174 (11.43)	126 (9.52)	485 (11.33)
Educational level (years)				
≤9	176 (12.28)	288 (18.92)	252 (19.03)	716 (16.73)
10–15	594 (41.45)	683 (44.88)	559 (42.22)	1836 (42.91)
≥16	663 (46.27)	551 (36.20)	513 (38.75)	1727 (40.36)
Pre-pregnancy BMI (kg/m^2^)				
<18.5	207 (14.45)	188 (12.35)	148 (11.17)	543 (12.69)
18.5–24	976 (68.10)	1068 (70.17)	950 (71.75)	2994 (69.97)
24–28	193 (13.47)	208 (13.67)	188 (14.20)	589 (13.77)
≥28	57 (3.98)	58 (3.81)	38 (2.87)	153 (3.58)
Sample collection season				
Spring/winter	617 (43.05)	660 (43.36)	552 (41.69)	1829 (42.74)
Summer/autumn	816 (56.94)	862 (56.64)	772 (58.31)	2450 (57.26)
Urinary iodine concentration (μg/L)				
<150	725 (50.59)	837 (54.99)	809 (61.10)	2371 (55.41)
150–249	385 (26.87)	407 (26.74)	318 (24.02)	1110 (25.94)
≥250	323 (22.54)	278 (18.26)	197 (14.88)	798 (18.65)

Values presented as the mean ± the standard deviation or N (%).

**Table 2 nutrients-14-04484-t002:** Thyroid hormone and antibodies in pregnant women under different vitamin D statuses during different pregnancy periods (median, 2.5th–97.5th).

Thyroid Hormones and Antibodies	The First Trimester		The Second Trimester		The Third Trimester	
	25(OH)D < 12 ng/mL	25(OH)D ≥ 12 ng/mL	25(OH)D < 12 ng/mL	25(OH)D ≥ 12 ng/mL	25(OH)D < 12 ng/mL	25(OH)D ≥ 12 ng/mL
TSH (mU/L)	1.29 (0.10–3.16)	1.17 (0.07–3.45)	1.62 (0.42–3.80)	1.62 (0.41–3.65)	1.82 (0.96–4.18)	1.77 (0.67–3.88)
TT3 (nmol/L)	1.91 (1.32–2.78)	1.96 (1.33–2.88)	2.21 (1.38–3.23)	2.21 (1.41–3.13)	2.14 (1.31–3.07)	2.18 (1.35–3.10)
TT4 (pmol/L)	125.13 (11.38–173.00)	125.40 (13.35–178.20)	130.15 (10.08–170.86)	125.54 (9.71–176.19)	123.40 (8.85–161.37)	119.30 (8.89–166.60)
FT3 (pg/mL)	5.55 (3.63–5.76)	4.53 (3.62–5.83)	4.24 (3.50–5.22)	4.16 (3.30–5.18)	3.92 (2.91–4.89)	3.86 (2.97–4.82)
FT4 (pmol/L)	**14.75 (9.98–19.57)**	**15.23 (9.91–20.03)**	13.23 (9.01–16.54)	12.95 (8.77–17.00)	11.78 (7.69–15.42)	12.02 (8.30–16.00)
TgAb (IU/mL)	10.00 (0.01–145.33)	10.00 (0.10–183.10)	10.00 (0.10–129.25)	10.00 (0.10–78.10)	10.00 (0.01–39.77)	10.00 (0.10–57.58)
TPOAb (IU/mL)	10.86 (0.23–59.15)	11.80 (0.30–103.44)	10.22 (0.29–57.77)	10.10 (0.29–63.30)	**10.82 (0.29–59.00)**	**9.30 (0.30–52.90)**
TrAb (IU/L)	0.30 (0.15–19.79)	0.30 (0.15–18.00)	0.30 (0.15–18.57)	0.30 (0.15–15.50)	0.30 (0.15–17.48)	0.30 (0.15–15.59)

Bold indicates statistical significance between different vitamin D statuses in the same pregnancy periods.

**Table 3 nutrients-14-04484-t003:** Interaction between low UIC and vitamin D deficiency on the risk of hypothyroidism/TgAb/TPOAb/TrAb positivity.

Vitamin D (ng/mL)	UIC: 150–249 μg/L		UIC < 150 μg/L		OR for UIC within Category of Vitamin D	RERI (95%CI)	AP (95%CI)
	Hypothyroidism/Normal (N)	OR (95%CI)	Hypothyroidism/Normal (N)	OR (95%CI)			
≥12	123/841	1.000	248/1821	1.09 (0.86, 1.38)	1.09 (0.86, 1.38)	−0.119 (−0.722, 0.484)	−0.154 (−0.955, 0.646)
<12	19/127	0.99 (0.59, 1.03)	31/271	0.76 (0.50, 1.16)	0.67 (0.41, 1.09)	-	-
OR for vitamin D deficiency within category of UIC	-	0.99 (0.59, 1.03)	-	0.84 (0.56, 1.24)	-	-	-
	TgAb positivity/Normal (N)	OR (95%CI)	TgAb positivity/Normal (N)	OR (95%CI)			
≥12	37/927	1.000	95/1974	0.78 (0.52, 1.15)	0.78 (0.52, 1.15)	0.932 (−0.006, 1.870)	0.600 (0.147, 1.054)
<12	2/144	0.32 (0.08, 1.37)	17/285	1.50 (0.83, 2.72)	0.23 (0.05, 1.03)	-	-
OR for vitamin D deficiency within category of UIC	-	0.32 (0.08, 1.37)	-	1.23 (0.72, 2.10)	-	-	-
	TPOAb positivity/Normal (N)	OR (95%CI)	TPOAb positivity/Normal (N)	OR (95%CI)			
≥12	188/776	1.000	410/1659	0.97 (0.80, 1.18)	0.97 (0.80, 1.18)	0.053 (−0.467, 0.573)	0.053 (−0.461, 0.567)
<12	26/120	0.92 (0.58, 1.45)	59/243	1.00 (0.72, 1.39)	0.88 (0.52, 1.50)	-	-
OR for vitamin D deficiency within category of UIC	-	0.92 (0.58, 1.45)	-	0.97 (0.71, 1.31)	-	-	-
	TrAb positivity/Normal (N)	OR (95%CI)	TrAb positivity/Normal (N)	OR (95%CI)			
≥12	201/763	1.000	452/1617	0.93 (0.77, 1.12)	0.93 (0.77, 1.12)	0.320 (−0.224, 0.864)	0.237 (−0.140, 0.613)
<12	29/117	0.93 (0.60, 1.44)	78/224	1.32 (0.97, 1.78)	0.67 (0.41, 1.09)	-	-
OR for vitamin D deficiency within category of UIC	-	0.93 (0.60, 1.44)	-	1.27 (0.96, 1.68)	-	-	-

**Table 4 nutrients-14-04484-t004:** Interaction between high UIC and vitamin D deficiency on the risk of hypothyroidism/TgAb/TpoAb/TrAb positivity.

Vitamin D(ng/mL)	UIC: 150–249 μg/L		UIC ≥ 249 μg/L		OR for UIC within Category of Vitamin D	RERI (95% CI)	AP (95% CI)
	Hypothyroidism/Normal (N)	OR (95% CI)	Hypothyroidism/Normal (N)	OR (95% CI)			
≥12	123/841	1.000	86/609	0.93 (0.69, 1.25)	0.93 (0.69, 1.25)	0.414 (−0.465, 1.293)	0.309 (−0.239, 0.857)
<12	19/127	0.99 (0.59, 1.67)	17/86	1.37 (0.78, 2.41)	1.89 (0.87, 4.11)	-	-
OR for vitamin D deficiency within category of UIC	-	0.99 (0.59, 1.67)	-	1.40 (0.79, 2.49)	-	-	-
	TgAb positivity/Normal (N)	OR (95% CI)	TgAb positivity/Normal (N)	OR (95% CI)			
≥12	37/927	1.000	33/662	1.17 (0.72, 1.89)	1.17 (0.72, 1.89)	0.644 (−0.580, 1.868)	0.558 (−0.179, 1.295)
<12	2/144	0.33 (0.08, 1.37)	5/98	1.13 (0.43, 2.98)	3.20 (0.60, 17.09)	-	-
OR for vitamin D deficiency within category of UIC	-	0.33 (0.08, 1.37)	-	1.02 (0.38, 2.69)	-	-	-
	TPOAb positivity/Normal (N)	OR (95% CI)	TPOAb positivity/Normal (N)	OR (95% CI)			
≥12	188/776	1.000	148/547	1.03 (0.81, 1.32)	1.03 (0.81, 1.32)	0.226 (−0.473, 0.925)	0.194 (−0.348, 0.736)
<12	26/120	0.92 (0.58, 1.45)	25/78	1.20 (0.74, 1.96)	1.33 (0.70, 2.54)	-	-
OR for vitamin D deficiency within category of UIC	-	0.92 (0.58, 1.45)	-	1.05 (0.99, 1.01)	-	-	-
	TrAb positivity/Normal (N)	OR (95% CI)	TrAb positivity/Normal (N)	OR (95% CI)			
≥12	201/763	1.000	134/561	0.87 (0.68, 1.12)	0.87 (0.68, 1.12)	**1.027 (0.162, 1.891)**	**0.559 (0.258, 0.861)**
<12	29/117	0.93 (0.60, 1.44)	34/69	**1.86 (1.12, 2.91)**	**2.10 (1.15, 3.83)**	-	-
OR for vitamin D deficiency within category of UIC	-	0.93 (0.60, 1.44)	-	**2.06 (1.31, 3.26)**	-	-	-

Bold indicates statistical significance after adjustment for age, BMI, annual household income, educational level, gestation week, season, smoking and drinking status, as well as vitamin D intake.

**Table 5 nutrients-14-04484-t005:** Thyroid hormone in pregnant women under different TrAb statuses during different pregnancy periods (median, 2.5th–97.5th).

Thyroid Hormones	The First Trimester		The Second Trimester		The Third Trimester	
	TrAb (−)	TrAb (+)	TrAb (−)	TrAb (+)	TrAb (−)	TrAb (+)
TSH (mU/L)	1.19 (0.08–3.51)	1.11 (0.06–3.06)	1.62 (0.42–3.71)	1.70 (0.37–3.42)	**1.80 (0.71–4.01)**	**1.65 (0.70–3.63)**
TT3 (nmol/L)	**1.98 (1.35–2.86)**	**1.84 (1.24–2.90)**	2.21 (1.39–3.17)	2.22 (1.43–3.06)	2.18 (1.34–3.15)	2.16 (1.38–2.97)
TT4 (pmol/L)	**124.00 (11.75–172.45)**	**130.85 (92.00–186.20)**	**124.00 (9.10–170.87)**	**134.15 (93.57–188.27)**	**117.40 (8.51–161.00)**	**129.80 (94.81–177.30)**
FT3 (pg/mL)	4.56 (3.63–5.83)	4.47 (3.58–5.66)	**4.19 (3.38–5.22)**	**4.13 (1.75–4.96)**	**3.91 (3.12–4.88)**	**3.77 (1.75–4.51)**
FT4 (pmol/L)	15.16 (10.89–19.99)	15.15 (9.06–19.86)	**12.72 (8.63–16.63)**	**13.99 (10.97–18.10)**	**11.63 (8.11–15.03)**	**13.20 (10.37–17.35)**

Bold indicates statistical significance between different TrAb statuses in the same pregnancy periods. TrAb (−): thyrotrophic receptor antibodies negative; TrAb (+): thyrotrophic receptor antibodies positive.

## Data Availability

The data presented in this study are available on request from the corresponding author. The data are not publicly available due to policy.

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
