# Peer review of "The Interactive Effects of Severe Vitamin D Deficiency and Iodine Nutrition Status on the Risk of Thyroid Disorder in Pregnant Women"

_nutrients, 2022, doi:10.3390/nu14214484_

Round 1
Reviewer 1 Report
The subject of the study is of interest to the readers of Nutrients. However, 1. authors must review their text extensively to correct mistakes concerning english language (tenses should be consistent either past or present, single or plural should be used).
2. There must be space between words, after parentheses, after commas or full stops to make reading easier.
3. Text in the results should preceed Tables and the relevant Tables should be referred to the appropriate Result section
4. At least Tables 3 and 4 must be simplified and made more clear. Also if possible to be restricted to one page each.
5. There is no need to refer to numbers and precentages in the discussion section (l203-208) as this information has already been given in the Tables and Result section.
Detailed corrections are shown as comments in the attached file.
4.

Reviewer 2 Report
Wei Lu, et al. studied the interactive effects of vitamin D and iodine on the risk of thyroid disorder in pregnant women. The authors showed severe vitamin D deficiency with excess iodine intake could increase the risk of trab positive in pregnant women. The interaction between vitamin D deficiency and iodine intake is interesting, but this study has some concerns.
1) There are several vitamin D metabolites. 25OHD concentrations can include inactive epimers. Just 25OHD concentrations cannot be enough to explore the associations. As epimers are associated with sun exposure and vitamin D supplementations, both should be included in the study at least.
2) The authors investigated the interaction between vitamin D deficiency and iodine intake using just crossover analysis. Sensitivity analysis by the stratified data will be required.
3) As the status of graves’ disease fluctuates depending on gestational weeks. Analysis should be performed according to the trimesters.
4) The effects of the interaction between vitamin D deficiency and iodine intake on thyroid functions should be described further in terms of the (molecular) mechanisms.
Round 2
Reviewer 2 Report
The authors have satisfactorily addressed all of my concerns.
Author Response
We greatly appreciated your valuable comments.